# Implicit Graph Neural Networks

**Fangda Gu[1]***
gfd18@berkeley.edu

**Heng Chang[2]***
changh17@mails.tsinghua.edu.cn

**Wenwu Zhu[3]**
wwzhu@tsinghua.edu.cn

**Somayeh Sojoudi[1,2]**
sojoudi@berkeley.edu

**Laurent El Ghaoui[1,2]**
elghaoui@berkeley.edu

[1]Department of Electrical Engineering and Computer Sciences, University of California at Berkeley
[2]Tsinghua-Berkeley Shenzhen Institute, Tsinghua University
[3]Department of Computer Science and Technology, Tsinghua University

## Abstract

Graph Neural Networks (GNNs) are widely used deep learning models that learn meaningful representations from graph-structured data. Due to the finite nature of the underlying recurrent structure, current GNN methods may struggle to capture long-range dependencies in underlying graphs. To overcome this difficulty, we propose a graph learning framework, called Implicit Graph Neural Networks (IGNN[2]), where predictions are based on the solution of a fixed-point equilibrium equation involving implicitly defined "state" vectors. We use the Perron-Frobenius theory to derive sufficient conditions that ensure well-posedness of the framework. Leveraging implicit differentiation, we derive a tractable projected gradient descent method to train the framework. Experiments on a comprehensive range of tasks show that IGNNs consistently capture long-range dependencies and outperform the state-of-the-art GNN models.

## 1 Introduction

Graph neural networks (GNNs) (Zhou et al., 2018; Zhang et al., 2020) have been widely used on graph-structured data to obtain a meaningful representation of nodes in the graph. By iteratively aggregating information from neighboring nodes, GNN models encode graph-relational information into the representation, which then benefits a wide range of tasks, including biochemical structure discovery (Gilmer et al., 2017; Wan et al., 2019), computer vision (Kampffmeyer et al., 2018), and recommender systems (Ying et al., 2018). Recently, newer convolutional GNN structures (Wu et al., 2019b) have drastically improved the performance of GNNs by employing various techniques, including renormalization (Kipf and Welling, 2016), attention (Veličković et al., 2017), and simpler activation (Wu et al., 2019a).

The aforementioned modern convolutional GNN models capture relation information up to $T$-hops away by performing $T$ iterations of graph convolutional aggregation. Such information gathering procedure is similar to forward-feeding schemes in popular deep learning models, such as multi-layer perceptron and convolutional neural networks. However, despite their simplicity, these computation strategies cannot discover the dependency with a range longer than $T$-hops away from any given node.

One approach tackling this problem is to develop recurrent GNNs that iterate graph convolutional aggregation until convergence, without any *a priori* limitation on the number of hops. This idea

arises in many traditional graph metrics, including eigenvector centrality (Newman, 2018) and PageRank (Page et al., 1999), where the metrics are implicitly defined by some fixed-point equation. Intuitively, the long-range dependency can be better captured by iterating the information passing procedure for an infinite number of times until convergence. Pioneered by (Gori et al., 2005), new recurrent GNNs leverage partial training (Gallicchio and Micheli, 2010, 2019) and approximation (Dai et al., 2018) to improve performance. With shared weights, these methods avoid exploding memory issues and achieve accuracies competitive with convolutional counterparts in certain cases.

While these methods offer an alternative to the popular convolutional GNN models with added benefits for certain problems, there are still significant limitations in evaluation and training for recurrent GNN models. Conservative convergence conditions and sophisticated training procedures have limited the use of these methods in practice, and outweighed the performance benefits of capturing the long-range dependency. In addition, most of these methods cannot leverage multi-graph information or adapt to heterogeneous network settings, as prevalent in social networks as well as bio-chemical graphs (Wan et al., 2019).

**Paper contributions.** In this work, we present the **I**mplicit **G**raph **N**eural **N**etwork (IGNN) framework to address the problem of evaluation and training for recurrent GNNs. We first analyze graph neural networks through a rigorous mathematical framework based on the Perron-Frobenius theory (Berman and Plemmons, 1994), in order to establish general well-posedness conditions for convergence. We show that most existing analyses are special cases of our result. As for training, we propose a novel projected gradient method to efficiently train the IGNN, where we leverage implicit differentiation methods to obtain the exact gradient, and use projection on a tractable convex set to guarantee well-posedness. We show that previous gradient methods for recurrent graph neural networks can be interpreted as an approximation to IGNN. Further, we extend IGNN to heterogeneous network settings. Finally, we conduct comprehensive comparisons with existing methods, and demonstrate that our method effectively captures long-range dependencies and outperforms the state-of-the-art GNN models on a wide range of tasks.

**Paper outline.** In Section 2, we give an overview of related work on GNN and implicit models. In Section 3, we introduce the background and notations for this paper. Section 4 discusses the IGNN framework together with its well-posedness and training under both ordinary and heterogeneous settings. Section 5 empirically compares IGNN with modern GNN methods.

## 2 Related Work

**GNN models.** Pioneered by (Gori et al., 2005), GNN models have gained influence for graph-related tasks. Led by GCN (Kipf and Welling, 2016), convolutional GNN models (Veličković et al., 2017; Hamilton et al., 2017; Wu et al., 2019a; Jin et al., 2019; Chang et al., 2020) involve a finite number of modified aggregation steps with different weight parameters. On the other hand, recurrent GNN models (Gori et al., 2005) use the same parameters for each aggregation step and potentially enable infinite steps. (Li et al., 2015) combines recurrent GNN with recurrent neural network structures. Methods such as Fast and Deep Graph Neural Network (FDGNN) (Gallicchio and Micheli, 2010, 2019) use untrained recurrent GNN models with novel initialization to its aggregation step for graph classification. While the Stochastic Steady-State Embedding (SSE) method (Dai et al., 2018) uses an efficient approximated training and evaluation procedure for node classification. Recently, global method Geom-GCN (Pei et al., 2020) employs additional embedding approaches to capture global information. However, Geom-GCN (Pei et al., 2020) also belongs to convolutional-based GNNs, which struggle to capture very long range dependency due to the finite iterations they take.

**Implicit Models.** Implicit models are emerging structures in deep learning where the outputs are determined implicitly by a solution of some underlying sub-problem. Recent works (Bai et al., 2019) demonstrate the potential of implicit models in sequence modeling, physical engine (de Avila Belbute-Peres et al., 2018) and many others (Chen et al., 2018; Amos et al., 2018).(El Ghaoui et al., 2020) proposes a general implicit framework with the prediction rule based on the solution of a fixed-point equilibrium equation and discusses the well-posedness of the implicit prediction rule.

**Oversmoothing.** To catch the long-range dependency, another intuitive approach is to construct deeper convolutional GNNs by stacking more layers. However, (Li et al., 2018) found that the learned

node embeddings become indistinguishable as the convolutional GNNs get deeper. This phenomenon is called *over-smoothing*. Since then, a line of empirical (Li et al., 2018; Chen et al., 2020; Rong et al., 2020) and theoretical (Oono and Suzuki, 2020; Zhao and Akoglu, 2020) works follows on the *over-smoothing* phenomenon. Unlike convolutional GNNs, IGNN adopts a different approach for long-range dependency based on recurrent GNNs and doesn't seem to suffer performance degradation as much even though it could be viewed as an infinite-layer GNN. See appendix E.5 for details.

## 3   Preliminaries

Graph neural networks take input data in the form of graphs. A graph is represented by $G = (V, E)$ where $V$ is the set of $n := |V|$ nodes (or vertices) and $E \subseteq V \times V$ is the set of edges. In practice, we construct an adjacency matrix $A \in \mathbb{R}^{n \times n}$ to represent the graph $G$: for any two nodes $i, j \in V$, if $(i, j) \in E$, then $A_{ij} = 1$; otherwise, $A_{ij} = 0$. Some data sets provide additional information about the nodes in the form of a feature matrix $U \in \mathbb{R}^{p \times n}$, in which the feature vector for node $i$ is given by $u_i \in \mathbb{R}^p$. When no additional feature information from nodes is provided, the data sets would require learning a feature matrix $U$ separately in practice.

Given graph data, graph models produce a prediction $\hat{Y}$ to match the true label $Y$ whose shape depends on the task. GNN models are effective in graph-structured data because they involve trainable aggregation steps that pass the information from each node to its neighboring nodes and then apply nonlinear activation. The aggregation step at iteration $t$ can be written as follows:

$$X^{(t+1)} = \phi(W^{(t)} X^{(t)} A + \Omega^{(t)} U), \tag{1}$$

where $X^{(t)} \in \mathbb{R}^{m \times n}$ stacks the state vectors of nodes in time step $t$ into a matrix, in which the state vector for node $i$ is denoted as $x^{(t)} \in \mathbb{R}^m$; $W^{(t)}$ and $\Omega^{(t)}$ are trainable weight matrices; $\phi$ an activation function. The state vectors $X^{(T)}$ at final iteration can be used as the representation for nodes that combine input features and graph spatial information. The prediction from the GNN models is given by $\hat{Y} = f_\Theta(X^{(T)})$, where $f_\Theta$ is some trainable function parameterized by $\Theta$. In practice, a linear $f_\Theta$ is often satisfactory.

Modern GNN approaches adopt different forms of graph convolution aggregation (1). Convolutional GNNs (Wu et al., 2019b) iterate (1) with $\Omega = 0$ and set $X^{(0)} = U$. Some works temper with the adjacency matrix using renormalization (Kipf and Welling, 2016) or attention (Veličković et al., 2017)). While recurrent GNNs use explicit input from features at each step with tied weights $W$ and $\Omega$, some methods replace the term $\Omega U$ with $\Omega_1 U A + \Omega_2 U$, in order to account for feature information from neighboring nodes (Dai et al., 2018). Our framework adopts a similar recurrent graph convolutional aggregation idea.

A heterogeneous network is an extended type of graph that contains different types of relations between nodes instead of only one type of edge. We continue to use $G = (V, \mathcal{E})$ to represent a heterogeneous network with the node set $V$ and the edge set $\mathcal{E} \subseteq V \times V \times R$, where $R$ is a set of $N := |R|$ relation types. Similarly, we define the adjacency matrices $A_i$, where $A_i$ is the adjacency matrix for relation type $i \in R$. Some heterogeneous networks also have relation-specific feature matrices $U_i$.

**Notation.** For a matrix $V \in \mathbb{R}^{p \times q}$, $|V|$ denotes its absolute value (*i.e.* $|V|_{ij} = |V_{ij}|$). The infinity norm, or the max-row-sum norm, writes $\|V\|_\infty$. The 1-norm, or the max-column-sum norm, is denoted as $\|V\|_1 = \|V^\top\|_\infty$. The 2-norm is shown as $\|V\|$ or $\|V\|_2$. We use $\otimes$ to represent the Kronecker product, $\langle \cdot, \cdot \rangle$ to represent inner product and use $\odot$ to represent component-wise multiplication between two matrices of the same shape. For a $p \times q$ matrix $V$, $\mathbf{vec}(V) \in \mathbb{R}^{pq}$ represents the vectorized form of $V$, obtained by stacking its columns (See Appendix A for details). According to the Perron-Frobenius theory (Berman and Plemmons, 1994), every squared non-negative matrix $M$ has a real non-negative eigenvalue that gives the largest modulus among all eigenvalues of $M$. This non-negative eigenvalue of $M$ is called the *Perron-Frobenius (PF) eigenvalue* and denoted by $\lambda_{\mathrm{pf}}(M)$ throughout the paper.

## 4   Implicit Graph Neural Networks

We now introduce a framework for graph neural networks called **I**mplicit **G**raph **N**eural **N**etworks (IGNN), which obtains a node representation through the fixed-point solution of a non-linear "equi-

librium" equation. The IGNN model is formally described by

$$\hat{Y} = f_\Theta(X), \tag{2a}$$
$$X = \phi(WXA + b_\Omega(U)). \tag{2b}$$

In equation (2), the input feature matrix $U \in \mathbb{R}^{p \times n}$ is passed through some affine transformation $b_\Omega(\cdot)$ parametrized by $\Omega$ (*i.e.* a linear transformation possibly offset by some bias). The representation, given as the "internal state" $X \in \mathbb{R}^{m \times n}$ in the rest of the paper, is obtained as the fixed-point solution of the equilibrium equation (2b), where $\phi$ preserves the same shape of input and output. The prediction rule (2a) computes the prediction $\hat{Y}$ by feeding the state $X$ through the output function $f_\Theta$. In practice, a linear map $f_\Theta(X) = \Theta X$ may be satisfactory.

Unlike most existing methods that iterate (1) for a finite number of steps, an IGNN seeks the fixed point of equation (2b) that is trained to give the desired representation for the task. Evaluation of fixed point can be regarded as iterating (1) for an infinite number of times to achieve a steady state. Thus, the final representation potentially contains information from all neighbors in the graph. In practice, this gives a better performance over the finite iterating variants by capturing the long-range dependency in the graph. Another notable benefit of the framework is that it is memory-efficient in the sense that it only maintains one current state $X$ without other intermediate representations.

Despite its notational simplicity, the IGNN model covers a wide range of variants, including their multi-layer formulations by stacking multiple equilibrium equations similar to (2b). The SSE (Dai et al., 2018) and FDGNN (Gallicchio and Micheli, 2019) models also fit within the IGNN formulation. We elaborate on this aspect in Appendix C.

IGNN models can generalize to heterogeneous networks with different adjacency matrices $A_i$ and input features $U_i$ for different relations. In that case, we have the parameters $W_i$ and $\Omega_i$ for each relation type $i \in R$ to capture the heterogenerity of the graph. A new equilibrium equation (3) is used:

$$X = \phi\left(\sum_i (W_i X A_i + b_{\Omega_i}(U_i))\right). \tag{3}$$

In general, there may not exist a unique solution for the equilibrium equation (2b) and (3). Thus, the notion of well-posedness comes into play.

## 4.1 Well-posedness of IGNNs

For the IGNN model to produce a valid representation, we need to obtain some *unique* internal state $X(U)$ given any input $U$ from equation (2b) for ordinary graph settings or equation (3) for heterogeneous network settings. However, the equilibrium equation (2b) and (3) can have no well-defined solution $X$ given some input $U$. We give a simple example in the scalar setting in Appendix B, where the solution to the equilibrium equation (2b) does not even exist.

In order to ensure the existence and uniqueness of the solution to equation (2b) and (3), we define the notion of *well-posedness* for equilibrium equations with activation $\phi$ for both ordinary graphs and hetergeneous networks. This notion has been introduced in (El Ghaoui et al., 2020) for ordinary implicit models.

**Definition 4.1** (Well-posedness on ordinary graphs). *The tuple $(W, A)$ of the weight matrix $W \in \mathbb{R}^{m \times m}$ and the adjacency matrix $A \in \mathbb{R}^{n \times n}$ is said to be well-posed for $\phi$ if for any $B \in \mathbb{R}^{m \times n}$, the solution $X \in \mathbb{R}^{m \times n}$ of the following equation*

$$X = \phi(WXA + B) \tag{4}$$

*exists and is unique.*

**Definition 4.2** (Well-posedness on heterogeneous networks). *The tuple $(W_i, A_i, i = 1, \dots, N)$ of the weight matrices $W_i \in \mathbb{R}^{m \times m}$ and the adjacency matrices $A_i \in \mathbb{R}^{n \times n}$ is said to be well-posed for $\phi$ if for any $B_i \in \mathbb{R}^{m \times n}$, the solution $X \in \mathbb{R}^{m \times n}$ of the following equation*

$$X = \phi\left(\sum_{i=1}^{N}(W_i X A_i + B_i)\right) \tag{5}$$

*exists and is unique.*

We first develop sufficient conditions for the well-posedness property to hold on ordinary graph settings with a single edge type. The idea is to limit the structure of $W$ and $A$ together to ensure well-posedness for a set of activation $\phi$.

In the following analysis, we assume that $\phi$ is component-wise non-expansive, which we refer to as the component-wise non-expansive (CONE) property. Most activation functions in deep learning satisfy the CONE property (*e.g.* Sigmoid, tanh, ReLU, Leaky ReLU, *etc.*). For simplicity, we assume that $\phi$ is differentiable.

We can now establish the following sufficient condition on $(W, A)$ for our model with a CONE activation to be well-posed. Our result hinges on the notion of Perron-Frobenius (PF) eigenvalue $\lambda_{\text{pf}}(M)$ for a non-negative matrix $M$, as well as the notion of Kronecker product $A \otimes B \in \mathbb{R}^{pm \times qn}$ between two matrices $A \in \mathbb{R}^{m \times n}$ and $B \in \mathbb{R}^{p \times q}$. See Appendix A for details.

**Theorem 4.1** (PF sufficient condition for well-posedness on ordinary graphs)**.** *Assume that $\phi$ is a component-wise non-expansive (CONE) activation map. Then, $(W, A)$ is well-posed for any such $\phi$ if $\lambda_{\text{pf}}(|A^\top \otimes W|) < 1$. Moreover, the solution $X$ of equation (4) can be obtained by iterating equation (4).*

*Proof.* Recall that for any three matrices $A, W, X$ of compatible sizes, we have $(A^\top \otimes W)\, \mathbf{vec}(X) = \mathbf{vec}(WXA)$ (Schacke, 2018). Showing equation (4) has an unique solution is equivalent to showing that the following "vectorized" equation has a unique solution:

$$\mathbf{vec}(X) = \phi(A^\top \otimes W\, \mathbf{vec}(X) + \mathbf{vec}(B))$$

It follows directly from Lemma B.1 that if $\lambda_{\text{pf}}(|A^\top \otimes W|) = \lambda_{\text{pf}}(A)\lambda_{\text{pf}}(|W|) < 1$, then the above equation has unique solution that can be obtained by iterating the equation. ∎

We find Theorem 4.1 so general that many familiar and interesting results will follow from it, as discussed in the following remarks. Detailed explanations can be found in Appendix B.

**Remark 4.1** (Contraction sufficient condition for well-posedness (Gori et al., 2005))**.** *For any component-wise non-expansive (CONE) $\phi$, if $\mathcal{A}(X) = \phi(WXA + B)$ is a contraction of $X$ (w.r.t. vectorized norms), then $(W, A)$ is well-posed for $\phi$.*

**Remark 4.2** (Well-posedness for directed acyclic graph)**.** *For a directed acyclic graph (DAG), let $A$ be its adjacency matrix. For any real squared $W$, it holds that $(W, A)$ is well-posed for every CONE activation map. Note that $\mathcal{A}(X) = \phi(WXA + B)$ need not be a contraction of $X$.*

**Remark 4.3** (Sufficient well-posedness condition for k-regular graph (Gallicchio and Micheli, 2019))**.** *For a k-regular graph, let $A$ be its adjacency matrix. $(W, A)$ is well-posed for every CONE activation map if $k\|W\|_2 < 1$.*

A similar sufficient condition for well-posedness holds for heterogeneous networks.

**Theorem 4.2** (PF sufficient condition for well-posedness on heterogeneous networks)**.** *Assume that $\phi$ is some component-wise non-expansive (CONE) activation map. Then, $(W_i, A_i,\ i = 1, \dots, N)$ is well-posed for any such $\phi$ if $\lambda_{\text{pf}}\left(\sum_{i=1}^N |A_i^\top \otimes W_i|\right) < 1$. Moreover, the solution $X$ of equation (5) can be obtained by iterating equation (5).*

We give a complete proof in Appendix B. Sufficient conditions in Theorems 4.1 and 4.2 guarantee convergence when iterating aggregation step to evaluate state $X$. Furthermore, these procedures enjoy exponential convergence in practice.

## 4.2 Tractable Well-posedness Condition for Training

At training time, however, it is difficult in general to ensure satisfaction of the PF sufficient condition $\lambda_{\text{pf}}(|W|)\lambda_{\text{pf}}(A) < 1$, because $\lambda_{\text{pf}}(|W|)$ is non-convex in $W$. To alleviate the problem, we give a numerically tractable convex condition for well-posedness that can be enforced at training time efficiently through projection. Instead of using $\lambda_{\text{pf}}(|W|) < \lambda_{\text{pf}}(A)^{-1}$, we enforce the stricter condition $\|W\|_\infty < \lambda_{\text{pf}}(A)^{-1}$, which guarantees the former inequality by $\lambda_{\text{pf}}(|W|) \le \|W\|_\infty$. Although $\|W\|_\infty < \lambda_{\text{pf}}(A)^{-1}$ is a stricter condition, we show in the following theorem that it is equivalent to the PF condition for positively homogeneous activation functions, (*i.e.* $\phi(\alpha x) = \alpha \phi(x)$ for any $\alpha \ge 0$ and $x$), in the sense that one can use the former condition at training without loss of generality.

**Theorem 4.3** (Rescaled IGNN). *Assume that $\phi$ is CONE and positively homogeneous. For an IGNN $(f_\Theta, W, A, b_\Omega, \phi)$ where $(W, A)$ satisfies the PF sufficient condition for well-posedness, namely $\lambda_{\mathrm{pf}}(|W|) < \lambda_{\mathrm{pf}}(A)^{-1}$, there exists a linearly-rescaled equivalent IGNN $(\tilde{f}_\Theta, W', A, \tilde{b}_\Omega, \phi)$ with $\|W'\|_\infty < \lambda_{\mathrm{pf}}(A)^{-1}$ that gives the same output $\hat{Y}$ as the original IGNN for any input U.*

The proof is given in Appendix B. The above-mentioned condition can be enforced by selecting a $\kappa \in [0, 1)$ and projecting the updated $W$ onto the convex constraint set $\mathcal{C} = \{W : \|W\|_\infty \leq \kappa/\lambda_{\mathrm{pf}}(A)\}$.

For heterogeneous network settings, we recall the following:

**Remark 4.4.** *For any non-negative adjacency matrix $A$ and arbitrary real parameter matrix $W$, it holds that $\|A^\top \otimes W\|_\infty = \|A^\top\|_\infty \|W\|_\infty = \|A\|_1 \|W\|_\infty$.*

Similar to the difficulty faced in the ordinary graph settings, ensuring the PF sufficient condition on heterogeneous networks is hard in general. We propose to enforce the following tractable condition that is convex in $W_i$'s: $\sum_{i=1}^N \|A_i\|_1 \|W_i\|_\infty \leq \kappa < 1$, $\kappa \in [0, 1)$. Note that this condition implies $\left\|\sum_{i=1}^N A_i^\top \otimes W_i\right\|_\infty \leq \kappa$, and thus $\lambda_{\mathrm{pf}}\left(\sum_{i=1}^N |A_i^\top \otimes W_i|\right) \leq \kappa < 1$. The PF sufficient condition for well-posedness on heterogeneous networks is then guaranteed.

## 4.3 Training of IGNN

We start by giving the training problem (6), where a loss $\mathcal{L}(Y, \hat{Y})$ is minimized to match $\hat{Y}$ to $Y$ and yet the tractable condition $\|W\|_\infty \leq \kappa/\lambda_{\mathrm{pf}}(A)$ for well-posedness is enforced with $\kappa \in [0, 1)$:

$$\min_{\Theta, W, \Omega} \mathcal{L}(Y, f_\Theta(X)) : \quad X = \phi(WXA + b_\Omega(U)), \quad \|W\|_\infty \leq \kappa/\lambda_{\mathrm{pf}}(A). \tag{6}$$

The problem can be solved by projected gradient descent (involving a projection to the well-posedness condition following a gradient step), where the gradient is obtained through an implicit differentiation scheme. From the chain rule, one can easily obtain $\nabla_\Theta \mathcal{L}$ for the parameter of $f_\Theta$ and $\nabla_X \mathcal{L}$ for the internal state $X$. In addition, we can write the gradient with respect to scalar $q \in W \cup \Omega$ as follows:

$$\nabla_q \mathcal{L} = \left\langle \frac{\partial (WXA + b_\Omega(U))}{\partial q}, \nabla_Z \mathcal{L} \right\rangle, \tag{7}$$

where $Z = WXA + b_\Omega(U)$ *assuming fixed* $X$ (see Appendix D). Here, $\nabla_Z \mathcal{L}$ is given as a solution to the equilibrium equation

$$\nabla_Z \mathcal{L} = D \odot \left(W^\top \nabla_Z \mathcal{L} \, A^\top + \nabla_X \mathcal{L}\right), \tag{8}$$

where $D = \phi'(WXA + b_\Omega(U))$ and $\phi'(z) = d\phi(z)/dz$ refers to the element-wise derivative of the CONE map $\phi$. Since $\phi$ is non-expansive, it is 1-Lipschitz (*i.e.* the absolute value of $d\phi(z)/dz$ is not greater than 1), the equilibrium equation (8) for gradient $\nabla_Z \mathcal{L}$ admits a *unique* solution by iterating (8) to convergence, if $(W, A)$ is well-posed for any CONE activation $\phi$. (Note that $D \odot (\cdot)$ can be seen as a CONE map with each entry of $D$ having absolute value less than or equal to 1.) Again, $\nabla_Z \mathcal{L}$ can be efficiently obtained due to exponential convergence when iterating (8) in practice.

Once $\nabla_Z \mathcal{L}$ is obtained, we can use the chain rule (via autograd software) to easily compute $\nabla_W \mathcal{L}$, $\nabla_\Omega \mathcal{L}$, and possibly $\nabla_U \mathcal{L}$ when input $U$ requires gradients (*e.g.* in cases of features learning or multi-layer formulation). The deriviation has a deep connection to the Implicit Function Theorm. See Appendix D for details.

Due to the norm constraint introduced for well-posedness, each update to $W$ requires a projection step (See Section 4.1). The new $W$ is given by $W^+ = \pi_\mathcal{C}(W) := \mathrm{argmin}_{\|M\|_\infty \leq \kappa/\lambda_{\mathrm{pf}}(A)} \|M - W\|_F^2$, where $\pi_\mathcal{C}$ is the projection back onto $\mathcal{C} = \{\|W\|_\infty \leq \kappa/\lambda_{\mathrm{pf}}(A)\}$. The projection is decomposible across the rows of $W$. Each sub-problem will be a projection onto an $\mathcal{L}_1$-ball for which efficient methods exist (Duchi et al., 2008). A similar projected gradient descent training scheme for heterogeneous network settings is detailed in Appendix D. Note that the gradient method in SSE (Dai et al., 2018) uses a first-order approximated solution to equation (8). FDGNN (Gallicchio and Micheli, 2019) only updates $\Theta$ at training using gradient descent.

# 5 Numerical Experiment

In this section, we demonstrate the capability of IGNN on effectively learning a representation that captures the long-range dependency and therefore offers the state-of-the-art performance on both synthetic and real-world data sets. More specifically, we test IGNN against a selected set of baselines on 6 node classification data sets (Chains, PPI, AMAZON, ACM, IMDB, DBLP) and 5 graph classification data sets (MUTAG, PTC, COX2, PROTEINS, NC11), where Chains is a synthetic data set; PPI and AMAZON are multi-label classification data sets; ACM, IMDB and DBLP are based on heterogeneous networks. We inherit the same experimental settings and reuse the results of baselines from literatures in some of the data sets. The test set performance is reported. Detailed description of the data sets, our preprocessing procedure, hyper-parameters, and other information of experiments can be found in Appendix E.

**Synthetic Chains Data Set.** To evaluate GNN's capability for capturing the underlying long-range dependency in graphs, we create the Chains data set where the goal is to classify nodes in a chain of length $l$. The information of the class is only sparsely provided as the feature in an end node. We use a small training set, validation set, and test set with only 20, 100, and 200 nodes, respectively. For simplicity, we only consider the binary classification task. Four representative baselines are implemented and compared. We show in Figure 1 that IGNN and SSE (Dai et al., 2018) both capture the long-range dependency with IGNN offering a better performance for longer chains, while finite-iterating convolutional GNNs with $T = 2$, including GCN (Kipf and Welling, 2016), SGC (Wu et al., 2019a) and GAT (Veličković et al., 2017), fail to give meaningful predictions when the chains become longer. However, selecting a larger $T$ for convolutional GNNs does not seem to help in this case of limited training data. We further discuss this aspect in Appendix E.

Figure 1: Micro-$F_1$ (%) performance with respect to the length of the chains.

Table 1: Multi-label node classification Micro-$F_1$ (%) performance on PPI data set.

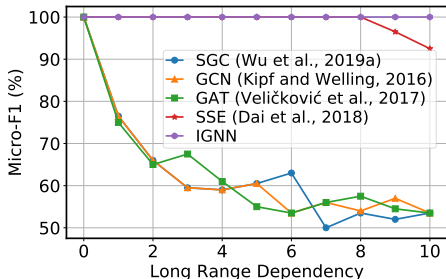

| Method | Micro-$F_1$ /% |
|---|---|
| Multi-Layer Perceptron | 46.2 |
| GCN (Kipf and Welling, 2016) | 59.2 |
| GraphSAGE (Hamilton et al., 2017) | 78.6 |
| SSE (Dai et al., 2018) | 83.6 |
| GAT (Veličković et al., 2017) | 97.3 |
| **IGNN** | **97.6** |

**Node Classification.** The popular benchmark data set Protein-Protein Interaction (PPI) models the interactions between proteins using a graph, with nodes being proteins and edges being interactions. Each protein can have at most 121 labels and be associated with additional 50-dimensional features. The train/valid/test split is consistent with GraphSage (Hamilton et al., 2017). We report the micro-averaged $F_1$ score of a multi-layer IGNN against other popular baseline models. The results can be found in Table 1. By capturing the underlying long-range dependency between proteins, the IGNN achieves the best performance compared to other baselines.

Figure 2: Micro/Macro-$F_1$ (%) performance on the multi-label node classification task with Amazon product co-purchasing network data set.

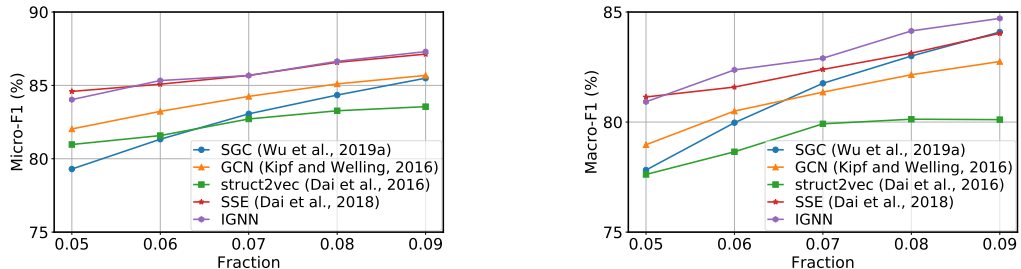

To further manifest the scalability of IGNN towards larger graphs, we conduct experiments on a large multi-label node classification data set, namely the Amazon product co-purchasing network data set (Yang and Leskovec, 2015) [3]. The data set renders products as nodes and co-purchases as edges but provides no input features. 58 product types with more than 5,000 products are selected from a total of 75,149 product types. While holding out 10% of the total nodes as test set, we vary the training set fraction from 5% to 9% to be consistent with (Dai et al., 2018). The data set come with no input feature vectors and thus require feature learning at training. Both Micro-$F_1$ and Macro-$F_1$ are reported on the held-out test set, where we compare IGNN with a set of baselines consistent with those in the synthetic data set. However, we use struct2vec (Dai et al., 2016) as an alternative to GAT since GAT faces a severe out-of-memory issue in this task.

As shown in Figure 2, IGNN again outperforms the baselines in most cases, especially when the amount of supervision grows. When more labels are available, more high-quality feature vectors of the nodes are learned and this enables the discovery of more long-range dependency. This phenomenon is aligned with our observation that IGNN achieves a better performance when there is more long-range dependency in the underlying graph.

**Graph Classification.** Aside from node classification, we test IGNN on graph classification tasks. A total of 5 bioinformatics benchmarks are chosen: MUTAG, PTC, COX2, NCI1 and PROTEINS (Yanardag and Vishwanathan, 2015). See details of data sets in Appendix E. Under the graph classification setting, we compare a multi-layer IGNN with a comprehensive set of baselines, including a variety of GNNs and a number of graph kernels. Following identical settings as (Yanardag and Vishwanathan, 2015; Xu et al., 2018), 10-fold cross-validation with LIB-SVM (Chang and Lin, 2011) is conducted. The average prediction accuracy and standard deviations are reported in Table 2. In this experiment, IGNN achieves the best performance in 4 out of 5 experiments given the competitive baselines. Such performance further validates IGNN's success in learning converging aggregation steps that capture long-range dependencies when generalized to unseen testing graphs.

Table 2: Graph classification accuracy (%). Results are averaged (and std are computed) on the outer 10 folds.

| Data sets | MUTAG | PTC | COX2 | PROTEINS | NCI1 |
|---|---|---|---|---|---|
| # graphs | 188 | 344 | 467 | 1113 | 4110 |
| Avg # nodes | 17.9 | 25.5 | 41.2 | 39.1 | 29.8 |
| DGCNN (Zhang et al., 2018) | 85.8 | 58.6 | − | 75.5 | 74.4 |
| DCNN (Atwood and Towsley, 2016) | 67.0 | 56.6 | − | 61.3 | 62.6 |
| GK (Shervashidze et al., 2009) | $81.4 \pm 1.7$ | $55.7 \pm 0.5$ | − | $71.4 \pm 0.3$ | $62.5 \pm 0.3$ |
| RW (Gärtner et al., 2003) | $79.2 \pm 2.1$ | $55.9 \pm 0.3$ | − | $59.6 \pm 0.1$ | − |
| PK (Neumann et al., 2016) | $76.0 \pm 2.7$ | $59.5 \pm 2.4$ | $81.0 \pm 0.2$ | $73.7 \pm 0.7$ | $82.5 \pm 0.5$ |
| WL (Shervashidze et al., 2011) | $84.1 \pm 1.9$ | $58.0 \pm 2.5$ | $83.2 \pm 0.2$ | $74.7 \pm 0.5$ | $\mathbf{84.5 \pm 0.5}$ |
| FDGNN (Gallicchio and Micheli, 2019) | $88.5 \pm 3.8$ | $63.4 \pm 5.4$ | $83.3 \pm 2.9$ | $76.8 \pm 2.9$ | $77.8 \pm 1.6$ |
| GCN (Kipf and Welling, 2016) | $85.6 \pm 5.8$ | $64.2 \pm 4.3$ | − | $76.0 \pm 3.2$ | $80.2 \pm 2.0$ |
| GIN (Xu et al., 2018) | $89.0 \pm 6.0$ | $63.7 \pm 8.2$ | − | $75.9 \pm 3.8$ | $82.7 \pm 1.6$ |
| IGNN | $\mathbf{89.3 \pm 6.7}$ | $\mathbf{70.1 \pm 5.6}$ | $\mathbf{86.9 \pm 4.0}$ | $\mathbf{77.7 \pm 3.4}$ | $80.5 \pm 1.9$ |

**Heterogeneous Networks.** Following our theoretical analysis on heterogeneous networks, we investigate how IGNN takes advantage of heterogeneity on node classification tasks. Three benchmarks based on heterogeneous network are chosen, *i.e.*, ACM, IMDB and DBLP (Wang et al., 2019; Park et al., 2019). More information regarding the heterogeneous network data sets can be found in Appendix E. Table 3 compares IGNN against a set of state-of-the-art GNN baselines for heterogeneous networks. The heterogeneous variant of IGNN continues to offer a competitive performance on all 3 data sets where IGNN gives the best performance in ACM and IMDB data sets. While on DBLP, IGNN underperforms DMGI but still outperforms other baselines by large margin. Good performance on heterogeneous networks demonstrates the flexibility of IGNN on handling heterogeneous relationships.

Table 3: Node classification Micro/Macro-$F_1$ (%) performance on heterogeneous network data sets.

| Data sets | ACM | | IMDB | | DBLP | |
|---|---|---|---|---|---|---|
| Metric | Micro-$F_1$ | Macro-$F_1$ | Micro-$F_1$ | Macro-$F_1$ | Micro-$F_1$ | Macro-$F_1$ |
| DGI (Veličković et al., 2018) | 88.1 | 88.1 | 60.6 | 59.8 | 72.0 | 72.3 |
| GCN/GAT | 87.0 | 86.9 | 61.1 | 60.3 | 71.7 | 73.4 |
| DeepWalk (Perozzi et al., 2014) | 74.8 | 73.9 | 55.0 | 53.2 | 53.7 | 53.3 |
| mGCN (Ma et al., 2019) | 86.0 | 85.8 | 63.0 | 62.3 | 71.3 | 72.5 |
| HAN (Wang et al., 2019) | 87.9 | 87.8 | 60.7 | 59.9 | 70.8 | 71.6 |
| DMGI (Park et al., 2019) | 89.8 | 89.8 | 64.8 | 64.8 | **76.6** | **77.1** |
| IGNN | **90.5** | **90.6** | **65.5** | **65.5** | 73.8 | 75.1 |

## 6 Conclusion

In this paper, we present the implicit graph neural network model, a framework of recurrent graph neural networks. We describe a sufficient condition for well-posedness based on the Perron-Frobenius theory and a projected gradient decent method for training. Similar to some other recurrent graph neural network models, implicit graph neural network captures the long-range dependency, but it carries the advantage further with a superior performance in a variety of tasks, through rigorous conditions for convergence and exact efficient gradient steps. More notably, the flexible framework extends to heterogeneous networks where it maintains its competitive performance.

## Broader Impact

GNN models are widely used on applications involving graph-structured data, including computer vision, recommender systems, and biochemical strucature discovery. Backed by more rigorous mathematical arguments, our research improves the capability GNNs of capturing the long-range dependency and therefore boosts the performance on these applications.

The improvements of performance in the applications will give rise to a better user experience of products and new discoveries in other research fields. But like any other deep learning models, GNNs runs into the problem of interpretability. The trade-off between performance and interpretability has been a topic of discussion. On one hand, the performance from GNNs benefits the tasks. On the other hand, the lack of interpretability might make it hard to recognize underlying bias when applying such algorithm to a new data set. Recent works (Hardt et al., 2016) propose to address the fairness issue by enforcing the fairness constraints.

While our research focuses on performance by capturing the long-range dependency, like many other GNNs, it does not directly tackle the fairness and interpretability aspect. We would encourage further work on fairness and interpretability on GNNs. Another contribution of our research is on the analysis of heterogeneous networks, where the fairness on treatment of different relationships remains unexplored. The risk of discrimination in particular real-world context might require cautious handling when researchers develop models.

## Acknowledgments and Disclosure of Funding

Funding in direct support of this work: National Key Research and Development Program of China (No. 2020AAA0107800, 2018AAA0102000), ONR Award N00014-18-1-2526, and other funding from Berkeley Artificial Intelligence Lab, Pacific Extreme Event Research Center, Genentech, Tsinghua-Berkeley Shenzhen Institute and National Natural Science Foundation of China Major Project (No. U1611461).

## Footnotes

*Equal contributions. Work done during Heng's visit to University of California at Berkeley.

[2]Code available at https://github.com/SwiftieH/IGNN.

[3] http://snap.stanford.edu/data/#amazon

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
