[Supplementary Material]

# Supplementary material

## A    Kronecker Product

For two matrices $A$ and $B$, the Kronecker product of $A \in \mathbb{R}^{m \times n}$ and $B \in \mathbb{R}^{p \times q}$ is denoted as $A \otimes B \in \mathbb{R}^{pm \times qn}$:

$$A \otimes B = \begin{pmatrix} A_{11}B & \cdots & A_{1n}B \\ \vdots & \ddots & \vdots \\ A_{m1}B & \cdots & A_{mn}B \end{pmatrix}.$$

By definition of the Kronecker product, $(A \otimes B)^\top = A^\top \otimes B^\top$. Additionally, the following equality holds assuming compatible shapes, $(A^\top \otimes W)\, \mathbf{vec}(X) = \mathbf{vec}(WXA)$ (Schacke, 2018), where $\mathbf{vec}(X) \in \mathbb{R}^{mn}$ denotes the vectorization of matrix $X \in \mathbb{R}^{m \times n}$ by stacking the columns of $X$ into a single column vector of dimension $mn$. Suppose $x_i \in \mathbb{R}^m$ is the $i$-th column of $X$, $\mathbf{vec}(X) = [x_1^\top, \ldots, x_n^\top]^\top$.

Leveraging the definition of Kronecker product and vectorization, the following equality holds, $(A^\top \otimes W)\, \mathbf{vec}(X) = \mathbf{vec}(WXA)$ (Schacke, 2018). Intuitively, this equality reshapes $WXA$ which is linear in $X$ into a more explicit form $(A^\top \otimes W)\, \mathbf{vec}(X)$ which is linear in $\mathbf{vec}(X)$, a flattened form of $X$. Through the transformation, we place $WXA$ into the form of $Mx$. Thus, we can employ Lemma B.1 to obtain the well-posedness conditions.

## B    Well-posedness of IGNN: Illustration, Remarks, and Proof

### B.1    A Scalar Example

Consider the following scalar equilibrium equation (9),

$$x = \mathrm{ReLU}(wxa + u), \tag{9}$$

where $x, w, a, u \in \mathbb{R}$ and $\mathrm{ReLU}(\cdot) = \max(\cdot, 0)$ is the rectified linear unit. If we set $w = a = 1$, the equation (9) will have no solutions for any $u > 0$. See Figure 3 for the example with $u = 1$.

Figure 3: Plots of $x$ (red plot) and $\mathrm{ReLU}(wxa + u) = \mathrm{ReLU}(x + 1)$ with $w = a = u = 1$ (blue plot). The two plots will intersect at some point whenever a solution exists. However in this case the two plots have no intersections, meaning that there is no solution to equation (9).

### B.2    Detailed Explanation for Remarks

**Remark B.1.** *For some non-negative adjacency matrix $A$, and arbitrary real parameter matrix $W$,* $\lambda_{\mathrm{pf}}(|A^\top \otimes W|) = \lambda_{\mathrm{pf}}(A^\top \otimes |W|) = \lambda_{\mathrm{pf}}(A)\lambda_{\mathrm{pf}}(|W|).$

The final equality of the above remark follows from the fact that, the spectrum of the Kronecker product of matrix $A$ and $B$ satisfies that $\Delta(A \otimes B) = \{\mu\lambda : \mu \in \Delta(A),\ \lambda \in \Delta(B)\}$, where $\Delta(A)$ represents the spectrum of matrix $A$. And that, the left and right eigenvalues of a matrix are the same.

We find Theorem 4.1 to be quite general. We show that many familiar and interesting results following from it.

**Remark B.2** (4.1, Contraction sufficient condition for well-posedness (Gori et al., 2005))**.** *For any component-wise non-expansive (CONE) $\phi$, if $\mathcal{A}(X) = \phi(WXA + B)$ is a contraction of $X$ (w.r.t. vectorized norms), then $(W, A)$ is well-posed for $\phi$.*

The above remark follows from the fact that the contraction condition for any CONE activation map is equivalent to $\|A^\top \otimes W\| < 1$, which implies $\lambda_{\mathrm{pf}}(|A^\top \otimes W|) < 1$.

**Remark B.3** (4.2, Well-posedness for directed acyclic graph)**.** *For a directed acyclic graph (DAG), let $A$ be its adjacency matrix. For any real squared $W$, we always have that $(W, A)$ is well-posed for any CONE activation map. Note that in this case $\mathcal{A}(X) = \phi(WXA + B)$ needs not be a contraction of $X$.*

Note that for DAG, $A$ is nilpotent ($\lambda_{\mathrm{pf}}(A) = 0$) and thus $\lambda_{\mathrm{pf}}(|A^\top \otimes W|) = \lambda_{\mathrm{pf}}(A)\lambda_{\mathrm{pf}}(|W|) = 0$.

**Remark B.4** (4.3, Sufficient well-posedness condition for k-regular graph (Gallicchio and Micheli, 2019))**.** *For a k-regular graph, let $A$ be its adjacency matrix. $(W, A)$ is well-posed for any CONE activation map if $k\|W\|_2 < 1$.*

It follows from that for a k-regular graph, the PF eigenvalue of the adjacency matrix $\lambda_{\mathrm{pf}}(A) = k$. And $\lambda_{\mathrm{pf}}(A)\lambda_{\mathrm{pf}}(|W|) \le k\|W\|_2 < 1$ guarantees well-posedness.

**Remark B.5** (4.4)**.** *For some non-negative adjacency matrix A, and arbitrary real parameter matrix $W$, $\|A^\top \otimes W\|_\infty = \|A^\top\|_\infty \|W\|_\infty = \|A\|_1 \|W\|_\infty$.*

The above remark follows from the facts that, $\|\cdot\|_\infty$ (resp. $\|\cdot\|_1$) gives maximum row (resp. column) sum of the absolute values of a given matrix. And that, for some real matrices $A$ and $B$, $\|A \otimes B\|_\infty = \max_{i,j}\left(\sum_{k,l} |A_{ik}B_{jl}|\right) = \max_{i,j}\left(\sum_k |A_{ik}| \sum_l |B_{jl}|\right) = \max_i\left(\sum_k |A_{ik}|\right) \max_j\left(\sum_l |B_{jl}|\right) = \|A\|_\infty \|B\|_\infty$.

## B.3 An Important Lemma for Well-posedness

**Lemma B.1.** *If $\phi$ is component-wise non-negative (CONE), $M$ is some squared matrix and $v$ is any real vector of compatible shape, the equation $x = \phi(Mx + v)$ has a unique solution if $\lambda_{\mathrm{pf}}(|M|) < 1$. And the solution can be obtained by iterating the equation. Hence, $x = \lim_{t\to\infty} x_t$.*

$$x_{t+1} = \phi(Mx_t + v), \; x_0 = 0, \; t = 0, 1, \ldots \tag{10}$$

*Proof.* For existence, since $\phi$ is component-wise and non-expansive, we have that for $t \ge 1$ and the sequence $x_0, x_1, x_2, \ldots$ generated from iteration (10),

$$|x_{t+1} - x_t| = |\phi(Mx_t + v) - \phi(Mx_{t-1} + v)| \le |M(x_t - x_{t-1})| \le |M||x_t - x_{t-1}|.$$

For $n > m \ge 1$, the following inequality follows,

$$|x_n - x_m| \le |M|^m \sum_{i=0}^{n-m-1} |M|^i |x_1 - x_0| \le |M|^m \sum_{i=0}^{\infty} |M|^i |x_1 - x_0| \le |M|^m w, \tag{11}$$

where

$$w := \sum_{i=0}^{\infty} |M|^i |x_1 - x_0| = (I - |M|)^{-1} |x_1 - x_0|.$$

Because $\lambda_{\mathrm{pf}}(|M|) < 1$, the inverse of $I - |M|$ exists. It also follows that $\lim_{t\to\infty} |M|^t = 0$. From inequality (11), we show that the sequence $x_0, x_1, x_2, \ldots$ is a Cauchy sequence because $0 \le \lim_{m\to\infty} |x_n - x_m| \le \lim_{m\to\infty} |M|^m w = 0$. And thus the sequence converges to some solution of $x = \phi(Mx + v)$.

For uniqueness, suppose both $x_a$ and $x_b$ satisfy $x = \phi(Mx + v)$, then the following inequality holds,

$$0 \le |x_a - x_b| \le |M||x_a - x_b| \le \lim_{t\to\infty} |M|^t |x_a - x_b| = 0.$$

It follows that $x_a = x_b$ and there exists unique solution to $x = \phi(Mx + v)$. ∎

### B.4 Proof of Theorem 4.2

*Proof.* Similarly, we can rewrite equation (5) into the following "vectorized" form.

$$\mathbf{vec}(X) = \phi\left(\sum_{i=1}^{N}(A_i^\top \otimes W_i)\,\mathbf{vec}(X) + \sum_{i=1}^{N}\mathbf{vec}(B_i)\right)$$

It follows from a similar scheme as the proof of Lemma B.1 that if $\lambda_{\mathrm{pf}}\left(\sum_{i=1}^{N}|A_i^\top \otimes W_i|\right) < 1$, the above equation has unique solution which can be obtained by iterating the equation. ∎

### B.5 Proof of Theorem 4.3

*Proof.* The proof is based on the following formula for PF eigenvalue (Berman and Plemmons, 1994).

$$\lambda_{\mathrm{pf}}(|W|) = \inf_{S}\ \|SWS^{-1}\|_\infty\ :\ S = \mathbf{diag}(s),\ \ s > 0 \tag{12}$$

In the case where $|W|$ has simple PF eigenvalue, problem (12) admits positive optimal scaling factor $s > 0$, a PF eigenvector of $|W|$. And we can design the equivalent IGNN $(\tilde{f}_\Theta, W', A, \tilde{b}_\Omega, \phi)$ with $\|W'\|_\infty < \lambda_{\mathrm{pf}}(A)^{-1}$ by rescaling:

$$\tilde{f}_\Theta(\cdot) = f_\Theta(S^{-1}\,\cdot),\quad W' = SWS^{-1},\quad \tilde{b}_\Omega(\cdot) = Sb_\Omega(\cdot),$$

where $S = \mathbf{diag}(s)$. ∎

## C Examples of IGNN

In this section we introduce some examples of the variation of IGNN.

**Multi-layer Setup.** It is straight forward to extend IGNN to a multi-layer setup with several sets of $W$ and $\Omega$ parameters for each layer. For conciseness, we use the ordinary graph setting. By treating the fixed-point solution $X_{l-1}$ of the $(l-1)$-th layer as the input $U_l$ to the $l$-th layer of equilibrium equation, a multi-layer formulation of IGNN with a total of $L$ layers is created.

$$
\begin{aligned}
\hat{Y} &= f_\Theta(X_L),\\
X_L &= \phi_L(W_L X_L A + b_{\Omega_L}(X_{L-1})),\\
&\ \vdots\\
X_l &= \phi_l(W_l X_l A + b_{\Omega_l}(X_{l-1})),\\
&\ \vdots\\
X_1 &= \phi_1(W_1 X_1 A + b_{\Omega_1}(U)),
\end{aligned}
\tag{13}
$$

where $\phi_1,\ldots,\phi_L$ are activation functions. We usually assume that CONE property holds on them. And $(W_l, \Omega_l)$ is the set of weights for the $l$-th layer. Thus the multi-layer formulation (13) with parameters $(W_l, l = 1,\ldots,L,\ A)$ is well-posed (*i.e.* gives unique prediction $\hat{Y}$ for any input $U$) when $(W_l, A)$ is well-posed for $\phi_l$ for any layer $l$. This is true since the well-posedness for a layer guarantees valid input for the next layer. Since all layers are well-posed, the formulation will give unique final output for any input of compatible shape. FDGNN (Gallicchio and Micheli, 2019) uses a similar multi-layer formulation for graph classification but is only partially trained in practive.

In terms of the affine input function, $b_\Omega(U) = \Omega U A$ is a good choice. We show that the multi-layer IGNN with such $b_\Omega$ is equivalent to a single layer IGNN (2) with higher dimensions, the same $A$ matrix and $f_\Theta$ function. The new activation map is given by $\phi = (\phi_L,\ldots,\phi_l,\ldots,\phi_1)$. Although $\phi$ is written in a block-wise form, they still operate on entry level and remain non-expansive. Thus the well-posedness results still hold. The new $\tilde{W}$ and $\tilde{b}_\Omega$ write,

$$
\tilde{W} = \begin{pmatrix} W_L & \Omega_L & & \\ & \ddots & \ddots & \\ & & \ddots & \Omega_2 \\ & & & W_1 \end{pmatrix},\quad \tilde{b}_\Omega(U) = \begin{pmatrix} 0 \\ \vdots \\ 0 \\ \Omega_1 \end{pmatrix} U A. \tag{14}
$$

**Special Cases.** Many existing GNN formulations including convolutional and recurrent GNNs can be treated as special cases of IGNN. We start by showing that GCN (Kipf and Welling, 2016), a typical example of convolutional GNNs, is indeed an IGNN. We give the matrix representation of a 2-layer GCN as follows,

$$\begin{aligned}\hat{Y} &= W_2 X_1 A, \\ X_1 &= \phi_1(W_1 U A),\end{aligned} \tag{15}$$

where $A$ is the renormalized adjacency matrix; $W_1$ and $W_2$ are weight parameters; $\phi_1$ is a CONE activation map for the first layer; and $X_1$ is the hidden representation of first layer. We show that GCN (15) is in fact a special case of IGNN by constructing an equivalent single layer IGNN (2) with the same $A$ matrix.

$$\hat{Y} = \tilde{f}_\Theta(\tilde{X}), \tag{16a}$$

$$\tilde{X} = \phi(\tilde{W}\tilde{X}A + \tilde{b}_\Omega(U)). \tag{16b}$$

The new state $\tilde{X} = (X_2, X_1)$. The new activation map is given by $\phi = (\phi_1, \mathbb{I})$, where $\mathbb{I}$ represents an identity map. And the new $\tilde{W}$, $\tilde{b}_\Omega$, and $\tilde{f}_\Theta(\tilde{X})$ are,

$$\tilde{W} = \begin{pmatrix} 0 & W_2 \\ 0 & 0 \end{pmatrix}, \quad \tilde{b}_\Omega(U) = \begin{pmatrix} 0 \\ W_1 \end{pmatrix} U A, \quad \tilde{f}_\Theta(\tilde{X}) = \begin{pmatrix} I \\ 0 \end{pmatrix} \tilde{X}. \tag{17}$$

This reformulation of single layer IGNN also extends to multi-layer GCNs with more than 2 layers as well as other convolutional GNNs. Note that the new $\tilde{W}$ for the equivalent single layer IGNN is always strictly upper triangular. Thus $|\tilde{W}|$ has only 0 eigenvalue. As a result, $\lambda_{\mathrm{pf}}(|A^\top \otimes W|) = \lambda_{\mathrm{pf}}(A)\lambda_{\mathrm{pf}}(|W|) = 0$ and the sufficient condition for well-posedness is always satisfied.

Another interesting special case is SSE (Dai et al., 2018), an example of recurrent GNN, that is given by

$$\begin{aligned}\hat{Y} &= W_2 X, \\ X &= \phi(W_{1r} W_2 X A + W_{1u} U A + W'_{1u} U),\end{aligned} \tag{18}$$

which can be easily converted into a single layer IGNN with the same $A$ matrix and CONE activation $\phi$. The new $\tilde{W}$, $\tilde{b}_\Omega$, and $\tilde{f}_\Theta(X)$ are,

$$\tilde{W} = W_{1r} W_2, \quad \tilde{b}_\Omega(U) = W_{1u} U A + W'_{1u} U, \quad \tilde{f}_\Theta(X) = W_2 X. \tag{19}$$

# D   Implicit differentiation for IGNN

To compute gradient of $\mathcal{L}$ from the training problem (6) w.r.t. a scalar $q \in W \cup \Omega$, we can use chain rule. It follows that,

$$\nabla_q \mathcal{L} = \left\langle \frac{\partial X}{\partial q}, \nabla_X \mathcal{L} \right\rangle, \tag{20}$$

where $\nabla_X \mathcal{L}$ can be easily calculated through modern autograd frameworks. But $\frac{\partial X}{\partial q}$ is non-trivial to obtain because $X$ is only implicitly defined. Fortunately, we can still leverage chain rule in this case by carefully taking the "implicitness" into account.

To avoid taking derivatives of matrices by matrices, we again introduce the vectorized representation $\mathbf{vec}(\cdot)$ of matrices. The vectorization of a matrix $X \in \mathbb{R}^{m \times n}$, denoted $\mathbf{vec}(X)$, is obtained by stacking the columns of $X$ into one single column vector of dimension $mn$. For simplicity, we use $\vec{X} := \mathbf{vec}(X)$ and $\nabla_{\vec{X}}\mathcal{L} = \mathbf{vec}(\nabla_X \mathcal{L})$ as a short hand notation of vectorization.

$$\frac{\partial \vec{X}}{\partial q} = \frac{\partial \vec{X}}{\partial \vec{Z}} \cdot \frac{\partial \vec{Z}}{\partial q}, \tag{21}$$

where $Z = WXA + b_\Omega(U)$ ($\vec{Z} = (A^\top \otimes W)\vec{X} + \overrightarrow{b_\Omega(U)}$) *assuming fixed $X$*. Unlike $X$ in equation (2b), $Z$ is not implicitly defined and should only be considered as a closed evaluation of $Z = WXA + b_\Omega(U)$ assuming $X$ doesn't change depending on $Z$. In some sense, the $Z$ in equation

(21) doesn't equal to $WXA + b_\Omega(U)$. However, the closeness property will greatly simplify the evaluation of $\frac{\partial \vec{Z}}{\partial q}$. It turns out that we can still employ chain rule in this case to calculate $\frac{\partial \vec{X}}{\partial \vec{Z}}$ for such $Z$ by taking the change of $X$ before hand into account as follows,

$$\frac{\partial \vec{X}}{\partial \vec{Z}} = \frac{\partial \phi(\vec{Z})}{\partial \vec{Z}} + \frac{\partial \phi\left((A^\top \otimes W)\vec{X} + \overrightarrow{b_\Omega(U)}\right)}{\partial \vec{X}} \cdot \frac{\partial \vec{X}}{\partial \vec{Z}}, \tag{22}$$

where the second term accounts for the change in $X$ that was ignored in $Z$. Another way to view this calculation is to right multiply $\frac{\partial \vec{Z}}{\partial q}$ on both sides of equation (22), which gives the chain rule evaluation of $\frac{\partial \vec{X}}{\partial q}$ that takes the gradient flowing back to $X$ into account:

$$\frac{\partial \vec{X}}{\partial q} = \frac{\partial \phi\left((A^\top \otimes W)\vec{X} + \overrightarrow{b_\Omega(U)}\right)}{\partial q} + \frac{\partial \phi\left((A^\top \otimes W)\vec{X} + \overrightarrow{b_\Omega(U)}\right)}{\partial \vec{X}} \cdot \frac{\partial \vec{X}}{\partial q}.$$

The equation (22) can be simplified as follows,

$$\frac{\partial \vec{X}}{\partial \vec{Z}} = (I - J)^{-1}\tilde{D}, \tag{23}$$

$$J = \frac{\partial \phi\left((A^\top \otimes W)\vec{X} + \overrightarrow{b_\Omega(U)}\right)}{\partial \vec{X}} = \tilde{D}(A^\top \otimes W),$$

where $\tilde{D} = \frac{\partial \phi(\vec{Z})}{\partial \vec{Z}} = \mathbf{diag}\left(\phi'\left((A^\top \otimes W)\vec{X} + \overrightarrow{b_\Omega(U)}\right)\right)$. Now we can rewrite equation (20) as

$$\nabla_q \mathcal{L} = \left\langle \frac{\partial \vec{Z}}{\partial q}, \nabla_{\vec{Z}}\mathcal{L} \right\rangle, \tag{24}$$

$$\nabla_{\vec{Z}}\mathcal{L} = \left(\frac{\partial \vec{X}}{\partial \vec{Z}}\right)^\top \nabla_{\vec{X}}\mathcal{L}, \tag{25}$$

which is equivalent to equation (7). $\nabla_{\vec{Z}}\mathcal{L}$ should be interpreted as the direction of steepest change of $\mathcal{L}$ for $Z = WXA + b_\Omega(U)$ *assuming fixed* $X$. Plugging equation (22) to (25), we arrive at the following equilibrium equation (equivalent to equation (8))

$$\nabla_{\vec{Z}}\mathcal{L} = \tilde{D}(A \otimes W^\top)\nabla_{\vec{Z}}\mathcal{L} + \tilde{D}\,\nabla_{\vec{X}}\mathcal{L},$$
$$\nabla_Z \mathcal{L} = D \odot \left(W^\top \nabla_Z \mathcal{L}\, A^\top + \nabla_X \mathcal{L}\right), \tag{26}$$

where $D = \phi'(WXA + b_\Omega(U))$. Interestingly, $\nabla_Z \mathcal{L}$ turns out to be given as a solution of an equilibrium equation particularly similar to equation (2b) in the IGNN "forward" pass. In fact, we can see element-wise multiplication with $D$ as a CONE "activation map" $\tilde{\phi}(\cdot) = D \odot (\cdot)$. And it follows from Section 4.1 that if $\lambda_{\mathrm{pf}}(W)\lambda_{\mathrm{pf}}(A) < 1$, then $\lambda_{\mathrm{pf}}(W^\top)\lambda_{\mathrm{pf}}(A^\top) < 1$ and $\nabla_Z \mathcal{L}$ can be uniquely determined by iterating the above equation (26). Although the proof will be more involved, if $(W, A)$ is well-posed for any CONE activation map, we can conclude that equilibrium equation (26) is also well-posed for $\tilde{\phi}$ where $\phi$ can be any CONE activation map.

Finally, by plugging the evaluated $\nabla_Z \mathcal{L}$ into equation (24), we get the desired gradients. Note that it is also possible to obtain gradient $\nabla_U \mathcal{L}$ by setting the $q$ in the above calculation to be $q \in U$. This is valid because we have no restrictions on selection of $q$ other than that it is not $X$, which is assumed fixed. Following the chain rule, we can give the closed form formula for $\nabla_W \mathcal{L}, \nabla_\omega \mathcal{L}, \omega \in \Omega$, and $\nabla_u \mathcal{L}, u \in U$.

$$\nabla_W \mathcal{L} = \nabla_Z \mathcal{L}\, A^\top X^\top, \quad \nabla_\omega \mathcal{L} = \left\langle \frac{\partial b_\Omega(U)}{\partial \omega}, \nabla_Z \mathcal{L} \right\rangle, \quad \nabla_u \mathcal{L} = \left\langle \frac{\partial b_\Omega(U)}{\partial u}, \nabla_Z \mathcal{L} \right\rangle.$$

**Heterogeneous Network Setting**  We start by giving the training problem for heterogeneous networks similar to training problem (6) for ordinary graphs,

$$\min_{\Theta, W, \Omega} \ \mathcal{L}(Y, f_\Theta(X))$$

$$\text{s.t.} \ \ X = \phi\left(\sum_{i=1}^{N}(W_i X A_i + b_{\Omega_i}(U_i))\right), \tag{27}$$

$$\sum_{i=1}^{N}\|A_i\|_1\|W_i\|_\infty \leq \kappa.$$

The training problem can be solved again using projected gradient descent method where the gradient of $W_i$ and $\Omega_i$ for $i \in R$ can be obtained with implicit differentiation. Using chain rule, we write the gradient of a scalar $q \in \bigcup_i(W_i \cup \Omega_i)$,

$$\nabla_q \mathcal{L} = \left\langle \frac{\partial\left(\sum_{i=1}^{N}(W_i X A_i + b_{\Omega_i}(U_i))\right)}{\partial q}, \nabla_Z \mathcal{L}\right\rangle, \tag{28}$$

where $Z = \sum_{i=1}^{N}(W_i X A_i + b_{\Omega_i}(U_i))$ and $\nabla_Z \mathcal{L}$ in equation (28) should be interpreted as "direction of fastest change of $\mathcal{L}$ for $Z$ *assuming fixed* $X$". Similar to the derivation in ordinary graphs setting, such notion of $\nabla_Z \mathcal{L}$ enables convenient calculation of $\nabla_q \mathcal{L}$. And the vectorized gradient w.r.t. $Z$ can be expressed as a function of the vectorized gradient w.r.t. $X$:

$$\nabla_{\vec{Z}} \mathcal{L} = \left(\frac{\partial \vec{X}}{\partial \vec{Z}}\right)^\top \nabla_{\vec{X}} \mathcal{L} \tag{29}$$

$$\frac{\partial \vec{X}}{\partial \vec{Z}} = \frac{\partial \phi(\vec{Z})}{\partial \vec{Z}} + \frac{\partial \phi\left(\sum_{i=1}^{N}\left((A_i^\top \otimes W_i)\vec{X} + \overrightarrow{b_{\Omega_i}(U_i)}\right)\right)}{\partial \vec{X}} \cdot \frac{\partial \vec{X}}{\partial \vec{Z}}$$

$$= (I - J)^{-1}\tilde{D} \tag{30}$$

$$J = \frac{\partial \phi\left(\sum_{i=1}^{N}\left((A_i^\top \otimes W_i)\vec{X} + \overrightarrow{b_{\Omega_i}(U_i)}\right)\right)}{\partial \vec{X}} = \tilde{D}\sum_{i=1}^{N}(A_i^\top \otimes W_i),$$

where $\tilde{D} = \frac{\partial \phi(\vec{Z})}{\partial \vec{Z}} = \mathbf{diag}\left(\phi'\left(\sum_{i=1}^{N}\left((A_i^\top \otimes W_i)\vec{X} + \overrightarrow{b_{\Omega_i}(U_i)}\right)\right)\right)$. Plugging the expression (30) into (29), we arrive at the following equilibrium equation for $\nabla_{\vec{Z}} \mathcal{L}$ and $\nabla_Z \mathcal{L}$,

$$\nabla_{\vec{Z}} \mathcal{L} = \tilde{D}\sum_{i=1}^{N}(A_i \otimes W_i^\top)\nabla_{\vec{Z}}\mathcal{L} + \tilde{D}\ \nabla_{\vec{X}}\mathcal{L}$$

$$\nabla_Z \mathcal{L} = D \odot \left(\sum_{i=1}^{N}(W_i^\top \nabla_Z \mathcal{L} A_i^\top) + \nabla_X \mathcal{L}\right), \tag{31}$$

where $D = \phi'\left(\sum_{i=1}^{N}(W_i X A_i + b_{\Omega_i}(U_i))\right)$. Not surprisingly, the equilibrium equation (31) again appears to be similar to the equation (3) in the IGNN "forward" pass. We can also view element-wise multiplication with $D$ as a CONE "activation map" $\tilde{\phi}(\cdot) = D \odot (\cdot)$. And it follows from Section 4.1 that if $\lambda_{\text{pf}}(|A^\top \otimes W|) < 1$, then $\lambda_{\text{pf}}(|A \otimes W^\top|) < 1$ and $\nabla_Z \mathcal{L}$ can be uniquely determined by iterating the above equation (26). It also holds that if $(W_i, A_i, i \in \{1, \dots, N\})$ is well-posed for any CONE activation $\phi$, then we can conclude that equilibrium equation (31) is also well-posed for $\tilde{\phi}$ where $\phi$ can be any CONE activation map.

Finally, by plugging the evaluated $\nabla_Z \mathcal{L}$ into equation (28), we get the desired gradients. It is also possible to obtain gradient $\nabla_{U_i} \mathcal{L}$ by setting the $q$ in the above calculation to be $q \in \bigcup_i U_i$. This is valid because we have no restrictions on selection of $q$ other than that it is not $X$, which is assumed fixed.

Figure 4: Chains with $l = 9$. Traditional methods fail even with more iterations.

After the gradient step, the projection to the tractable condition mentioned in Section 4.2 can be done approximately by assigning $\kappa_i$ for each relation $i \in R$ and projecting $W_i$ onto $\mathcal{C}_i = \{\|W_i\|_\infty \leq \kappa_i / \|A\|_1\}$. Ensuring $\sum_i \kappa_i = \kappa < 1$ will guarantee that the PF condition for heterogeneous network is satisfied. However, empirically, setting $\kappa_i < 1$ with $\sum_i \kappa_i > 1$ in some cases is enough for the convergence property to hold for the equilibrium equations.

## E    More on Experiments

In this section, we give detailed information about the experiments we conduct.

For preprocessing, we apply the *renormalization trick* consistent with GCN (Kipf and Welling, 2016) on the adjacent matrix of all data sets.

In terms of hyperparameters, unless otherwise specified, for IGNN, we use affine transformation $b_\Omega(U) = \Omega U A$; linear output function $f_\Theta(X) = \Theta X$; ReLU activation $\phi(\cdot) = \max(\cdot, 0)$; learning rate $0.01$; dropout with parameter $0.5$ before the output function; and $\kappa = 0.95$. We tune layers, hidden nodes, and $\kappa$ through grid search. The hyperparameters for other baselines are consistent with that reported in their papers. Results with identical experimental settings are reused from previous works.

### E.1    Synthetic Chains Data Set

We construct a synthetic node classification task to test the capability of models of learning to gather information from distant nodes. We consider the chains directed from one end to the other end with length $l$ (*i.e.* $l + 1$ nodes in the chain). For simplicity, we consider binary classification task with 2 types of chains. Information about the type is only encoded as 1/0 in first dimension of the feature (100d) on the starting end of the chain. The labels are provided as one-hot vectors (2d). In the data set we choose chain length $l = 9$ and 20 chains for each class with a total of 400 nodes. The training set consists of 20 data points randomly picked from these nodes in the total 40 chains. Respectively, the validation set and test set have 100 and 200 nodes.

A single-layer IGNN is implemented with 16 hidden unites and weight decay of parameter $5 \times 10^{-4}$ for all chains data sets with different $l$. Four representative baselines are chosen: Stochastic Steady-state Embedding (**SSE**) (Dai et al., 2018), Graph Convolutional Network (**GCN**) (Kipf and Welling, 2016), Simple Graph Convolution (**SGC**) (Wu et al., 2019a) and Graph Attention Network (**GAT**) (Veličković et al., 2017). They all use the same hidden units and weight decay as IGNN. For (**GAT**), 8 head attention is used. For (**SSE**), we use the embedding directly as output and fix-point iteration $n_h = 8$, as suggested (Dai et al., 2018).

As mentioned in Section 5, convolutional GNNs with $T = 2$ cannot capture the dependency with a range larger than 2-hops. To see how convolutional GNNs capture the long-range dependency as $T$ grows, we give an illustration of Micro-$F_1$ verses $T$ for the selected baselines in Figure 4. From the experiment, we find that convolutional GNNs cannot capture the long-range dependency given larger $T$. This might be a result of the limited number of training nodes in this chain task. As $T$ grows, convolutional GNNs experience an explosion of number of parameters to train. Thus the training data becomes insufficient for these models as the number of parameters increases.

Table 4: The overview of data set statistics in node classification tasks.

| Data set | # Nodes | # Edges | # Labels | Label type | Graph type |
|---|---|---|---|---|---|
| **Amazon (transductive)** | 334,863 | 925,872 | 58 | Product type | Co-purchasing |
| **PPI (inductive)** | 56,944 | 818,716 | 121 | Bio-states | Protein |

## E.2 Node Classification

For node classification task, we consider the applications under both transductive (Amazon) (Yang and Leskovec, 2015) and inductive (PPI) (Hamilton et al., 2017) settings. Transducive setting is where the model has access to the feature vectors of all nodes during training, while inductive setting is where the graphs for testing remain completely unobserved during training. The statistics of the data sets can be found in Table 4.

For experiments on Amazon, we construct a one-layer IGNN with 128 hidden units. No weight decay is utilized. The hyper parameters of baselines are consistent with (Yang and Leskovec, 2015; Dai et al., 2018).

For experiments on PPI, a five-layer IGNN model is applied for this multi-label classification tasks with hidden units as [1024, 512, 512, 256, 121] and $\kappa = 0.98$ for each layer. In addition, four MLPs are applied between the first four consecutive IGNN layers. We use the identity output function. Neither weight decay nor dropout is employed. We keep the experimental settings of baselines consistent with (Veličković et al., 2017; Dai et al., 2018; Kipf and Welling, 2016; Hamilton et al., 2017).

## E.3 Graph Classification

For graph classification, 5 bioinformatics data sets are employed with information given in Table 2. We compare IGNN with a comprehensive set of baselines, including a variety of GNNs: Deep Graph Convolutional Neural Network (**DGCNN**) (Zhang et al., 2018), Diffusion-Convolutional Neural Networks (**DCNN**) (Atwood and Towsley, 2016), Fast and Deep Graph Neural Network (**FDGNN**) (Gallicchio and Micheli, 2019), **GCN** (Kipf and Welling, 2016) and Graph Isomorphism Network (**GIN**) (Xu et al., 2018), and a number of state-of-the-art graph kernels: Graphlet Kernel (**GK**) (Shervashidze et al., 2009), Random-walk Kernel (**RW**) (Gärtner et al., 2003), Propagation Kernel (**PK**) (Neumann et al., 2016) and Weisfeiler-Lehman Kernel (**WL**) (Shervashidze et al., 2011). We reuse the results from literatures (Xu et al., 2018; Gallicchio and Micheli, 2019) since the same experimental settings are maintained.

As of IGNN, a three-layer IGNN is constructed for comparison with the hidden units of each layer as 32 and $\kappa = 0.98$ for all layers. We use an MLP as the output function. Besides, batch normalization is applied on each hidden layer. Neither weight decay nor dropout is utilized.

## E.4 Heterogeneous Networks

For heterogeneous networks, three data sets are chosen (ACM, IMDB, and DBLP). Consistent with previous works (Park et al., 2019), we use the the publicly available ACM data set (Wang et al., 2019), preprocessed DBLP and IMDB data sets (Park et al., 2019). For ACM and DBLP data sets, the nodes are papers and the aim is to classify the papers into three classes (Database, Wireless Communication, Data Mining), and four classes (DM, AI, CV, NLP)[4], respectively. For IMDB data set, the nodes are movies and we aim to classify these movies into three classes (Action, Comedy, Drama). The detailed information of data sets can be referred to Table 5. The preprocessing procedure and splitting method on three data sets keep consistent with (Park et al., 2019).

State-of-the-art baselines are selected for comparison with IGNN, including no-attribute network embedding: **DeepWalk** (Perozzi et al., 2014), attributed network embedding: **GCN**, **GAT** and **DGI** (Veličković et al., 2018), and attributed multiplex network embedding: **mGCN** (Ma et al., 2019), **HAN** (Wang et al., 2019) and **DMGI** (Park et al., 2019). Given the same experimental settings, we reuse the results of baselines from (Park et al., 2019).

Table 5: Statistics of the data sets for heterogeneous graphs (Park et al., 2019). The node attributes are bag-of-words of text. Num. labeled data denotes the number of nodes involved during training.

| | Relations (A-B) | Num. A | Num. B | Num. A-B | Relation type | Num. relations | Num. node attributes | Num. labeled data | Num. classes |
|---|---|---|---|---|---|---|---|---|---|
| ACM | Paper-Author | 3,025 | 5,835 | 9,744 | P-A-P | 29,281 | 1,830 (Paper abstract) | 600 | 3 |
| | Paper-Subject | 3,025 | 56 | 3,025 | P-S-P | 2,210,761 | | | |
| IMDB | Movie-Actor | 3,550 | 4,441 | 10,650 | M-A-M | 66,428 | 1,007 (Movie plot) | 300 | 3 |
| | Movie-Director | 3,550 | 1,726 | 3,550 | M-D-M | 13,788 | | | |
| DBLP | Paper-Author | 7,907 | 1,960 | 14,238 | P-A-P | 144,783 | 2,000 (Paper abstract) | 80 | 4 |
| | Paper-Paper | 7,907 | 7,907 | 10,522 | P-P-P | 90,145 | | | |
| | Author-Term | 1,960 | 1,975 | 57,269 | P-A-T-A-P | 57,137,515 | | | |

A one-layer IGNN with hidden units as 64 is implemented on all data sets. Similar to **DMGI**, a weight decay of parameter 0.001 is used. For ACM, $\kappa = (0.55, 0.55)$ is used for Paper-Author and Paper-Subject relations. For IMDB, we select $\kappa = (0.5, 0.5)$ for Movie-Actor and Movie-Director relations. For DBLP, $\kappa = (0.7, 0.4)$ is employed for Paper-Author and Paper-Paper relations. As mentioned in Appendix D, in practice, the convergence property can still hold when $\sum_i \kappa_i > 1$.

### E.5  Over-smoothness

Convolutional GNNs has suffered from over-smoothness when the model gets deep. An interesting question to ask is whether IGNN suffers from the same issue and experience performance degradation in capturing long-range dependency with its "infinitively deep" GNN design.

In an effort to answer this question, we compared IGNN against two latest convolutional GNN models that solve the over-smoothness issue, GCNII Chen et al. (2020) and DropEdge Rong et al. (2020). We use the same experimental setting as the Chains experiment in section 5. Both GCNII and DropEdge are implemented with 10-layer and is compared with IGNN in capturing long-range dependency. The result is reported in Figure 5. We observe that IGNN consistently outperforms both GCNII and DropEdge as the chains gets longer. The empirical result suggest little suffering from over-smoothness for recurrent GNNs.

Figure 5: Micro-$F_1$ (%) performance with respect to the length of the chains.

## Footnotes

[4]**DM**: KDD,WSDM,ICDM, **AI**: ICML,AAAI,IJCAI, **CV**: CVPR, **NLP**: ACL,NAACL,EMNLP