[Reviews · NeurIPS 2020]

Review 1

Summary and Contributions: The authors propose an implicit graph neural network (IGNN) to capture long-range dependencies in graphs. The proposed model is based on a fixed-point equilibrium equation. The authors first use the Perron-Frobenius theory to derive the well-posedness conditions of the model. Then, they introduce a trackable projection gradient descent method to effectively train the model. The authors evaluate their model on different data sets in different settings to demonstrate the ability of the model to capture long-range dependencies.

Strengths: • The paper studies the ability of the GNN models to capture long-range dependencies, which is a very important and difficult problem. The authors give a novel methodology based on a fixed-point equilibrium equation. • The authors provide a very reasonable theoretical analysis, and derive the well-posedness conditions of model convergence based on the Perron-Frobenius theory. • The authors verify the effectiveness of the proposed algorithm, the projected gradient method, through a series of experiments. The experiments on the synthetic chains data set can well illustrate the capabilities of the method to capture the underlying long-range dependences in graphs.

Weaknesses: • In section 4.3, the authors propose a projected gradient method to train the IGNN. But the paper lacks a discussion on the complexity of the model, and the training cost or the convergence process is not shown or analyzed in the experiment, which makes me concern about applicability of the model in practice. • Another minor concern is that for the node classification task, the proposed model is only compared with a few methods, while ignoring other advanced methods mentioned in the paper, such as FDGNN and GIN.

Correctness: The claims and solutions in the paper are technically sounded. The empirical methodology is thought out and conniving, which supports the claims well.

Clarity: The paper is clearly written and well organized.

Relation to Prior Work: Authors discuss the limitation of existing GNN models such as GCN, FDGNN and SSE. The proposed model is novel and different from previous contributions.

Reproducibility: Yes

Additional Feedback:


Review 2

Summary and Contributions: The paper addresses the difficulty of convolutional GNNs to capture long-range dependencies in the graph due to their finite fixed number of iterations regardless of the convergence. To tackle this problem the authors turn to recurrent architectures while proposing an implicit graph learning framework, where the state vectors of a recurrent architecture are implicitly defined, and optimization is done over equilibrium states simulating an infinite recurrence, which overcomes some of the disadvantages of recurrence architectures. The authors provide a wide theoretical analysis of their approach introducing well posedness conditions. To satisfy these conditions, the optimization of the model is done using the projected gradient method.

Strengths: The paper provides a full theoretical support to the claims. The main contributions: 1. Deriving the well posedeness conditions for their framework. 2. Deriving the implicit differentiation for the fixed point equation. 3. Incorporating projected gradient into the optimization process.

Weaknesses: --- Empirical evaluation --- The empirical evaluation section is not sufficient in my opinion, for two main reasons: 1. Claiming that the model is able to capture long range dependencies, I would have expected to see comparison to other global graph methods such as [1] which have their code online. 2. Graph classification results - all improvements in SOTA are within the std of the results. It is a known problem with those small and noisy datasets, which do not provide a real assessment of the method. I would like to see experiments on larger datasets. -- Discussion -- A more elaborated discussion of the computational complexity is missing, especially in the projected gradient part. [1] Pei, H., Wei, B., Chang, K.C., Lei, Y., & Yang, B. (2020). Geom-GCN: Geometric Graph Convolutional Networks. ArXiv, abs/2002.05287.

Correctness: Both theoretical and empirical methods are good.

Clarity: The paper is clear and well written.

Relation to Prior Work: Yes.

Reproducibility: Yes

Additional Feedback: ---- POST REBUTTAL ---- Decided to keep my score, however, with low confidence in my assessment.


Review 3

Summary and Contributions: The paper proposes a graph neural network called Implicit Graph Neural Networks. The proposed method exploits the implicit function theorem to estimate gradients towards the equilibrium state of iterative procedure. Constrained optimization methods, projected GD is employed to facilitate training. Experiments evaluated on graph classification and node tasks.

Strengths: It is interesting to learn the equilibrium of GNN iteration. It is good that authors are trying to analyze conditions where the method would fail and succeed.

Weaknesses: After reviewer discussions and reading rebuttal, I strongly believe this paper should not be accepted at NeurIPS. 1) Concern on over-smoothing and sub-optimality of the method is not addressed. 2) Concern on convergence is detrimental for this paper (as shared by R2) a) The paper claims convergence, but this is never demonstrated empirically. I asked for empirical support but authors fail to respond. b) The theoretical assumption for convergence rely on strong assumption and do not hold for common GNN architectures. 3) Clarity and reproducibility are of serious concern (as shared by R4). The description of the algorithm is unclear and important details are missing. This makes the paper not reproducible and the results are questionable. I assume reproducibility is a very important issue for NeurIPS. 4) Many state of the art baselines are missing. This shows concern for whether the method is actually good in performance. The author response results (Fig. 1) show they do not outperform previous methods. Not to mention the paper is not reproducible, so results are questionable. 5) Assignment of credit is unfair. The paper simply builds upon implicit function theorem but never explicitly discuss that. Moreover, related work on such equilibrium models are missing. 6) Many other points. ---------------------------------------- 1. The motivation is not convincing. It's not compelling that GNN should converge to an equilibrium. In fact, literature show that deeper GNNs often suffer from 'over-smoothing', i.e., learning too much noise from large neighborhood and perform worse than shallow GNN. Implicit GNN, which tries to iterate until converge, would also suffer from over-smoothing. [1]-[4] address this issue and make deeper GNNs also generalize well. Unfortunately, authors fail to discuss and compare with this line of works. 2. The proposed GNN architecture is rather weak, with only 1 linear layer plus phi. The conditions for uniqueness of equilibrium also relies on this weak form of architecture. Analysis for more reasonable architecture is desired. 3. Clarity is of concern. The description of the algorithm and how the chain rule is applied to estimate gradient is confusing. The implicit function theorem is not defined. line 172, Perron-Frobenius (PF) eigenvalue is never defined. Line 230, q is not defined. Line 233, phi(z)' never used anywhere in the algorithm. (8) algorithm is not clearly defined. 4. Error analysis for gradient estimation and runtime is not provided. 5. Motivation of heterogeneous graphs is not compelling. It's not clear how the proposed method can benefit heterogeneous graphs compared to other methods. 6. State-of-the-art baselines [5] for experimental evaluation is missing. 7. It's not clear how helpful the projected GD is. Ablation study is desired. 8. Convergence is not demonstrated empirically. Can authors show that implicit GNN indeed converge to equilibrium? References [1] Representation Learning on Graphs with Jumping Knowledge Networks. ICML 2018. [2] Predict then Propagate: Graph Neural Networks meet Personalized PageRank. ICLR 2019. [3] DropEdge: Towards Deep Graph Convolutional Networks on Node Classification. ICLR 2020. [4] Simple and Deep Graph Convolutional Networks. ICML 2020. [5] Graph Neural Tangent Kernel: Fusing Graph Neural Networks with Graph Kernels. NeurIPS 2019.

Correctness: The correctness is difficult o evaluate because the description of the algorithm is not clear.

Clarity: Background and prior art are missing, making it difficult to general audience to follow. For example, line 172, Perron-Frobenius (PF) eigenvalue is never defined. The implicit function theorem is not introduced or defined. Line 230, q is not defined. Line 233, phi(z)' never used anywhere in the algorithm. (8) algorithm is not clearly defined.

Relation to Prior Work: Several important literature are missing. Authors should be properly cite and discuss these papers [1]-[5]. A line of works study show that deeper GNNs often suffer from 'over-smoothing', i.e., learning too much noise from large neighborhood and perform worse than shallow GNN. [1]-[4] address this issue and make deeper GNNs also generalize well. Implicit GNN would also suffer from over-smoothing. Unfortunately, authors fail to cite and compare with these works. [1] Representation Learning on Graphs with Jumping Knowledge Networks. ICML 2018. [2] Predict then Propagate: Graph Neural Networks meet Personalized PageRank. ICLR 2019. [3] DropEdge: Towards Deep Graph Convolutional Networks on Node Classification. ICLR 2020. [4] Simple and Deep Graph Convolutional Networks. ICML 2020. The graph classification results miss state-of-the-art [5] which perform better than the implicit GNN, but authors fail to cite and compare. [5] Graph Neural Tangent Kernel: Fusing Graph Neural Networks with Graph Kernels. NeurIPS 2019.

Reproducibility: No

Additional Feedback: I suggest authors to make the writing clearer, and properly discuss related work and benchmarks.


Review 4

Summary and Contributions: Recurrent graph neural networks effectively capture the long-range dependency among nodes, however face the limitation of conservative convergence conditions and sophisticated training procedure. To address this problem, this paper proposes the implicit graph neural network framework (IGNN). While existing recurrent graph neural networks require to propagate a large number of steps to reach steady states during training, the proposed method only requires one step. This is realized by projected gradient descent. While generalizing several existing methods, the proposed method is more efficient to reach convergence. == After rebuttal: As pointed out the another reviewer, the paper indeed can be improved by having more discussion on over-smoothing, providing theoretical proof on convergence, include more reproducibility details and more experiments.

Strengths: Novelty: fair. This paper adopts the project gradient descent method to solve the equilibrium equation of message passing. Condition for convergence is more rigorous, and the training procedure is much simpler than previous recurrent graph neural networks. Empirical evaluation: comprehensive. This paper validates the effectiveness of its proposed method through a wide variety of experiments, including simulated dataset, node classification, graph classification, and heterogeneous graphs. Relevance to the NeurIPS community: very relevant. It is of interest to researchers about how to achieve node steady states for graphs in an efficient and effective way.

Weaknesses: Reproducibility: not good. It is better to provide algorithms for the training procedure and the evaluation procedure to make things clear. Besides, source codes are not available. Empirical evaluation: lack of analysis. The proposed method does not outperform baseline methods on some of the datasets including NC1 and DBLP. It is better to analyze the reason behind it. Limitation: it does not handle edge features.

Correctness: Seems correct.

Clarity: Yes.

Relation to Prior Work: The paper explains the difference in a few sentences. Perhaps it is better to discuss it in more detail

Reproducibility: No

Additional Feedback: 1. In training stage, do you only propagate node information only once and then calculate the gradients? While in evaluation stage, do you propagate node information for a finite number of time steps? 2. For equation 16, what is the results of \tilde{W}X ? It is better to write down the derivation process clearly to let the readers fully understand how GCN is a special cases of IGNN. 3. What pooling operations do you use in the graph classification experiments? Why SSE is not included in the graph classification task? 4 Can you provide a comparative analysis about training efficiency between SSE and IGNN? 5. Is your graph assumed to be directed or undirected?

[Author Response · NeurIPS 2020]

We thank all reviewers for the comments and the following response will be reflected in the final version.

**Complexity** (Rev1234): In solving equilibrium equation (2b), (3) for fixed-point state and (8) for gradient, we iterate the equations and the iterations converge when the well-posedness condition is satisfied as we mention in line 199 and 238. At training and test time, the equilibrium equations are iterated to convergence until the 2-norm difference between the LHS and RHS is less then some threshold epsilon. The convergence is guaranteed by well-posedness theory introduced in Section 4.1. In fact, the convergence is exponential both in theory and in practice. In terms of the projection step after a gradient update, projection onto $\|W\|_\infty \leq \kappa < 1$ ball can be decomposed over rows with each row $w_i$ given by a projection onto $\|w_i\|_1 \leq \kappa$, for which a straight forward $O(n \log n)$ algorithm exists using bisection. Duchi et al. (2008) has proposed an $O(n)$ algorithm for projection onto $L_1$ ball as we mention in line 246. We will offer a subsection with detailed complexity analysis and comparison with other methods in the final version.

**FDGNN and GIN not in node classification** (Rev1): FDGNN and GIN have mainly discussed their applications in graph classification. Thus we focus on the comparison in the graph classification task. More experiments will be added.

**More experiments with Geom-GCN and larger graphs** (Rev2): Please find the experimental results in Figure 1 and Table 1. Global methods like Geom-GCN employ additional embedding approaches to capture global information. However, convolutional-GNN-based methods struggle to capture very long range dependency due to the finite iterations they take. Geom-GCN is no exception. We also add a graph classification comparison on a larger and less noisy graph dataset COLLAB where IGNN continues to achieve the best performance.

**IGNN not well motivated to find equilibrium** (Rev3): We strongly disagree with the criticism from reviewer 3. IGNN and other recurrent GNN models including the first GNNs (Gori et al., 2005) are all based on the idea of seeking the equilibrium in the graph. Such idea further roots from traditional graph algorithms and metrics including eigenvector centrality (Newman, 2010) PageRank (Page et al., 1999), collaborate filtering as bipartite graph (Zhou, 2018) and more. **IGNN will suffer from 'over-smoothness'. Missing discussion.** (Rev3): We do not explore the direction because our work builds on recurrent GNN which is fundamentally different from convolutional GNN that suffers from 'over-smoothness' problem. IGNN obviously does not suffer from 'over-smoothness' as reflected from experiments where the 'infinitely deep' IGNN even outperforms a variety of models on a range of tasks. Additional experiments show that latest deep

Figure 1: Micro-$F_1$ (%) performance w.r.t. the length of the chains. Same experimental setting as that for Figure 1. We use 10 layers for GCNII and DropEdge.

models (DropEdge [3] and GCNII [4]) proposed by the reviewer that solve 'over-smoothness' cannot match the IGNN's performance in capturing long range dependency (See Figure 1). **IGNN is weak with 1 linear layer plus phi** (Rev3): The proposed architecture is not weak —- it covers a 100-layer GCN as a special case and many other models. The concise notation allows to formulate those in a way that looks like it has only one layer. See Section 4 and Appendix C for details. **PF eigenvalue (line 172) not defined.** (Rev3): Please find the definition from line 115 in the Preliminary. **State-of-the-art baseline [5] perform better than IGNN** (Rev3): Wrong. Due the space limit, we report performance on 4 graph classification tasks in Table 1 and IGNN outperforms [5] on all of them. We will add the results to the final version. **Does IGNN indeed converge to equilibrium? Why is projected GD needed?** (Rev3): Please find the well-posedness theorems in Section 4.1 which prove convergence of IGNNs that satisfy the well-posedness condition. And indeed it converges in practice. The projection step detailed in Section 4.2 is the essential procedure to enforce such well-posedness condition. We strongly encourage Rev3 to read Section 4 for better understanding.

**Underperforming NC1 and DBLP** (Rev4): For NCI1, IGNN ranks the second best among the all GNN variants, which is very competitive too. We believe the reason is, though GNNs learn high quality embedding, they can still underperform in distinguishing non-isomorphic (sub-)graphs compared with graph kernels (WL as the best performer). For DBLP, IGNN achieves the second best performance (after DMGI) using only 2 relationships out of 3 to be consistent with our settings on the other two datasets.

Table 1: Graph classification accuracy (%). Results are averaged (and std are computed) on the outer 10 folds.

| Data sets | PTC | COX2 | PROTEINS | COLLAB |
|---|---|---|---|---|
| WL | $58.0 \pm 2.5$ | $83.2 \pm 0.2$ | $74.7 \pm 0.5$ | $78.9 \pm 1.9$ |
| GIN | $63.7 \pm 8.2$ | $-$ | $75.9 \pm 3.8$ | $80.1 \pm 1.9$ |
| GNTK[5] | $67.9 \pm 6.9$ | $84.4 \pm 3.7$ | $75.6 \pm 4.2$ | $83.6 \pm 1.0$ |
| IGNN | $\mathbf{70.1 \pm 5.6}$ | $\mathbf{86.9 \pm 4.0}$ | $\mathbf{77.7 \pm 3.4}$ | $\mathbf{84.6 \pm 2.0}$ |

**Additional details** (Rev4): Though we use undirected graphs in the experiments, IGNN is not restricted to undirected graphs. For graph classification, we use mean pooling. Since SSE mainly discusses on learning node embedding and node classification in their paper, we would like to focus the comparison with SSE on node classification. Edge features are highly interesting direction to look at for IGNN. We will try to extend IGNN for it. For 2-layer GCN (15), $\tilde{W}X = [0, W_2; 0, 0][X_2; X_1] = [W_2X_1; 0]$. We will update the draft accordingly for better illustration.

[Meta-Review · NeurIPS 2020]

The authors propose an implicit graph neural network (IGNN) to capture long-range dependencies in graphs. The proposed model is based on a fixed-point equilibrium equation. The authors first use the Perron-Frobenius theory to derive the well-posedness conditions of the model. Then, they introduce a trackable projection gradient descent method to effectively train the model. The authors evaluate their model on different data sets in different settings to demonstrate the ability of the model to capture long-range dependencies. The paper is suggested for publication.